

# Sialyltransferase-related genes as predictive factors for therapeutic response and prognosis in cervical cancer

Jia Shao[1], Can Zhang[1], Yaonan Tang[1], Aiqin He[1] and Xiangyan Cheng[2]

[1] Department of Gynecology Oncology, Affiliated Tumor Hospital of Nantong University, Nantong, China
[2] Department of Obstetrics and Gynecology, Nantong Third People's Hospital, Affiliated Nantong Hospital 3 of Nantong University, Nantong, China

## ABSTRACT

**Background:** Cancer-associated hypersialylation is believed to be related to the metastatic cell phenotype and the suppression of sialyltransferases (SiaTs) has been suggested to be a potent preventive strategy against metastasis. The present research discovered SiaTs-related genes for cervical cancer (CC).

**Methods:** The Cancer Genome Atlas (TCGA) and Gene Expression Omnibus (GEO) databases were applied to obtain the relevant samples. Mutation dataset were processed using mutect2 software. The gene modules were obtained *via* weighted gene co-expression network analysis (WGCNA), and the enrichment analysis on the genes within the modules was implemented. Cox regression analysis and "glmnet" R package were applied to establish the relevant risk model. "MCPcounter" R package, ESTIMATE algorithm and TIMER online tools were used to depict the tumor immune microenvironment in CC. The mutation landscape was additionally plotted, and the response to immunotherapy in different cohorts were compared. Further reverse-transcription quantitative PCR and Transwell assays were performed to verify the expression and potential function of the screened key genes.

**Results:** Mutation of 14 SiaTs was seen in CC. Subsequently, WGCNA-based identification of SiaTs-related gene modules was significantly enriched in metabolism-related pathways. The established RiskScore model manifested excellent prognostic classification efficiency. A poorer prognosis and occurrence of both immune evasion and reduced immunoreactivity may be seen in high-risk patients yet relatively higher immune cell scores were noticeable in low-risk patients. Angiogenesis and MYC target V2 may be the differentially activated pathways in high-risk patients, while those in low-risk patients were KRAS Signaling DN and Interferon alpha response. In addition, most immune checkpoint-correlated genes were identified to express higher in low-risk patients, while higher sensitivities to chemotherapy drugs was discovered in high-risk patients. Cellular assays have revealed that *KCNK15*, *LIF*, *TCN2*, *SERPINF2*, and *CXCL3* were highly expressed yet *PIH1D2*, *DTX1* and *GCNT2* were low-expressed in Hela cells and that silencing *CXCL3* diminished the number of migrated and invaded Hela cells.

**Conclusion:** In this study, we systematically constructed and validated a risk scoring model based on SiaTs-related genes, which can effectively predict the prognosis and potential response to immunotherapy and chemotherapy in CC patients. This

Corresponding author
Xiangyan Cheng,
15962956278@163.com

provides a new molecular basis and clinical reference for achieving individualized treatment.

# INTRODUCTION

Cervical cancer (CC) currently is a prevalent female malignancy around the world both in terms of incidence and mortality, with over 500,000 cases diagnosed and around 300,000 mortalities each year (*Nisha et al., 2023*; *Wang et al., 2023*). At present, with the well characterization on the progression of CC, persistent human papillomavirus (HPV) infection is recognized as the central cause due to its effect on infecting metaplastic cells at the cervical transformation zone and integrating into the host genome (*Nisha et al., 2023*; *Shiri Aghbash et al., 2023*; *Yan et al., 2023*). The approach to treatment is contingent on the extent of the disease at the point of diagnosis and may include surgical interventions for early-stage cases, as well as adjuvant therapies or combination treatments for those with locally advanced disease. Survival forecasts primarily rely on the stages defined by the International Federation of Gynecology and Obstetrics. For patients experiencing recurrence or metastasis (stage IV), the anticipated 5-year survival rate remains low, at 15% or less. Regrettably, there are limited effective treatment options specifically targeting recurrent or metastatic cervical cancer. Notably, immunotherapies show potential benefits for these individuals. Hence, it is crucial to urgently identify new biological markers and promising therapeutic targets, as well as to establish effective immunotherapy approaches in clinical practice.

A large percentage of cancer-related mortalities have been identified to be the cause of metastasis. Despite the already gained insights into the migratory and invasive abilities of malignant cells, a detailed examination and understanding on the genomic basis of these processes, the role of glycosylation in particular, should be incorporated and considered (*Dobie & Skropeta, 2021*). Glycosylation refers to a post-translational modification (PTM) occurring in the endoplasmic reticulum and Golgi apparatus, and aberrant glycosylation has been already demonstrated to be associated with diverse processes during oncogenesis, like tumor growth, metastasis, chemoresistance and tumor immunity (*Reily et al., 2019*; *Hugonnet et al., 2021*). When it comes to cancer cells, hypersialylation has been established as one of the commonly discovered and altered glycosylation changes, which refers to the increase on the density of sialoglycans (the glycans containing sialic acid) (*Boligan et al., 2015*). A total of 20 different sialyltransferases (SiaTs) have been identified in humans, which are implicated in glycan synthesis and manifest peculiar features and preferences, with significantly varied expression levels between different tumors or even within the tumors of the same origin (*Haas, Simillion & Von Gunten, 2018*; *Haas et al., 2019*). The past decades have witnessed a profound increased number of research being conducted to evaluate the role of SiaTs in diverse malignancies like ovarian cancer, prostate cancer, and

pancreatic cancer, to name a few (*Wu et al., 2018*; *Scott et al., 2023*; *Bhalerao et al., 2023*). Notably, a SiaTs-related gene signature has been applied to estimate the therapeutic responses and prognostic results for bladder cancer patients (*Cao et al., 2023*).

This study uniquely combines big data bioinformatics analysis with cellular experiments to systematically reveal for the first time the critical role of SiaTs-related genes in the prognostic assessment and prediction of therapeutic response in CC. We first obtained CC patient sample data based on public databases and utilized weighted gene co- expression network analysis (WGCNA) in order to identify gene modules associated with SiaTS. Subsequently, univariate Cox and LASSO regression analyses were performed to screen for prognosis-related genes and construct a risk score model. In addition, we assessed immune cell infiltration, gene mutation status, and explored their sensitivity to immunotherapy and chemotherapeutic agents in different risk groups. Finally, the expression and biological functions of the key markers screened by this study were verified using cell-based experiments. This study not only extends the understanding of SiaTS-related genes in CC, but also provides new understanding into the prognostic evaluation and clinical management of CC patients.

## METHODS

### Data collection and pre-processing

The data of somatic mutation and clinical phenotype in CC were obtained and downloaded from The Cancer Genome Atlas (TCGA) database. The samples without survival time information and survival status were removed while retaining those with the survival time >0 day. Relevant RNA-seq expression profile was also downloaded and converted to transcripts per million (TPM) format, followed by the log2 transformation. The final sample of 291 CC patients to be used for subsequent analyses and information on the clinical characteristics of all patients is shown in Table 1. GSE44001 dataset with survival information in CC was downloaded from Gene Expression Omnibus (GEO, https://www.ncbi.nlm.nih.gov/geo/query/acc.cgi?acc=GSE44001) (*Lee et al., 2013*). The probes were converted into Symbol as per the annotation file, and the samples without overall survival status or clinical follow-up data were also removed. 300 CC samples were then sorted and applied in the analysis. In addition, the 20 SiaTs were all downloaded based on the existing study (*Cao et al., 2023*).

### Weighted gene co-expression analysis (WGCNA)

The specific gene modules were analyzed and identified *via* "WGCNA" R package (*Langfelder & Horvath, 2008*), and all processes mentioned were based on a study published elsewhere (*Jia et al., 2024*). Specifically, the soft threshold for WGCNA was determined with the help of "pickSoftThreshold" function, and then the gene modules were identified *via* hierarchical clustering based on the criteria of at least 60 genes per module (minModuleSize = 60). "GSVA" R package to be used to calculate SiaTs scores. Next, genes in the module with relatively higher correlation were applied for subsequent analyses (*Wang et al., 2024*).

**Table 1 Clinicopathological information of train and test cohorts.**

| Characteristics | Train cohort (N = 204) | Test cohort (N = 87) | Total (N = 291) | p-value | FDR |
|---|---|---|---|---|---|
| Age | | | | | |
| Mean ± SD | 48.36 ± 14.31 | 47.47 ± 12.61 | 48.09 ± 13.81 | | |
| Median (min-max) | 47.00 (20.00, 88.00) | 45.00 (25.00, 85.00) | 46.00 (20.00, 88.00) | | |
| T.stage | | | | 0.63 | 1 |
| T1 | 96 (32.99%) | 41 (14.09%) | 137 (47.08%) | | |
| T2 | 45 (15.46%) | 22 (7.56%) | 67 (23.02%) | | |
| T3 | 10 (3.44%) | 6 (2.06%) | 16 (5.50%) | | |
| T4 | 6 (2.06%) | 4 (1.37%) | 10 (3.44%) | | |
| N.stage | | | | 0.01 | 0.08 |
| N0 | 98 (33.68%) | 30 (10.31%) | 128 (43.99%) | | |
| N1 | 30 (10.31%) | 25 (8.59%) | 55 (18.90%) | | |
| M.stage | | | | 0.63 | 1 |
| M0 | 73 (25.09%) | 34 (11.68%) | 107 (36.77%) | | |
| M1 | 6 (2.06%) | 4 (1.37%) | 10 (3.44%) | | |
| Stage | | | | 0.63 | 1 |
| I | 115 (39.52%) | 44 (15.12%) | 159 (54.64%) | | |
| II | 44 (15.12%) | 20 (6.87%) | 64 (21.99%) | | |
| III | 25 (8.59%) | 16 (5.50%) | 41 (14.09%) | | |
| IV | 15 (5.15%) | 6 (2.06%) | 21 (7.22%) | | |
| Grade | | | | 0.57 | 1 |
| G1 | 14 (4.81%) | 4 (1.37%) | 18 (6.19%) | | |
| G2 | 92 (31.62%) | 37 (12.71%) | 129 (44.33%) | | |
| G3 | 76 (26.12%) | 40 (13.75%) | 116 (39.86%) | | |
| G4 | 1 (0.34%) | 0 (0.0e+0%) | 1 (0.34%) | | |
| Status | | | | 1 | 1 |
| Alive | 154 (52.92%) | 66 (22.68%) | 220 (75.60%) | | |
| Dead | 50 (17.18%) | 21 (7.22%) | 71 (24.40%) | | |

## Enrichment analysis

The selected genes within the specific module were subjected to Gene Ontology (GO) and Kyoto Encyclopedia of Genes and Genomes (KEGG) enrichment analyses employing "clusterProfiler" R package at the sorting threshold of $p < 0.05$ (*Yu et al., 2012*; *Xu et al., 2024*).

## Prognostic model construction and verification

Univariate Cox regression analysis was utilized to determine the association of the selected genes with the prognosis, and then the "glmnet" R package was applied for LASSO Cox regression analysis to narrow down the number of selected genes (*Engebretsen & Bohlin, 2019*). Multivariate Cox regression analysis was hereafter commenced for the generation of

risk coefficient of each identified gene and the following formula was applied for the calculation of RiskScore:

$$\text{RiskScore} = \Sigma\beta i \times \text{Expi}$$

(Note: "β" refers to the corresponding regression coefficient of identified genes and "Expi" means the expression of identified genes).

The optimal threshold was then calculated *via* "survminer" R package to allocate patients into high- or low-risk group (*Liu et al., 2021*). Thereafter, the Kaplan-Meier (KM) survival analysis with logarithmic rank test was applied to analyze the survival of patients, and the efficacy of prognostic model was evaluated *via* plotting the receiver operator characteristic (ROC) curves and calculating the corresponding area under the curve (AUC) values.

## Tumor immune microenvironment (TIME) analysis

The scores of 10 cell populations were calculated *via* the "MCPcounter" R package (*Zheng et al., 2021*). The overall TIME infiltration score was evaluated using the ESTIMATE algorithm. In addition, the relative abundance of six cell types was determined *via* TIMER online tool (http://cistrome.org/TIMER).

## SiaTs mutation landscape

The SiaTs mutation landscape and the gene mutation landscape in different risk groups were plotted based on the datasets processed by mutect2 software from TCGA.

## Immuno/chemotherapy response and drug sensitivity analyses

TIDE website (http://tide.dfci.harvard.edu/) was employed to compute the TIDE score and the degree of anti-tumor immune evasion in patients receiving immunotherapy was then predicted. Higher TIDE score was indicative of higher possibility of immune evasion and lower benefit from immunotherapy. Also, relevant immune checkpoint genes were acquired from the HisgAtlas database (https://bio.tools/HisgAtlas) and their levels in different risk groups were compared.

The drug sensitivity in patients of different risk groups in TCGA-CESC was predicted and compared based on the estimated half maximal inhibitory concentration (IC50) value using the "pRRophetic" R package (*Geeleher, Cox & Huang, 2014*).

## Cell culture and transfection

Both human cervical endometrial epithelial cells (HCerEpiC, CP-H058) and CC cell line Hela (CL-0101) commercially obtained from Procell (Wuhan, China) were cultured in minimal essential medium (PM150410; Procell, Wuhan, China) with the supplementation of 10% bovine calf serum (164210; Procell, Wuhan, China) and 1% penicillin-streptomycin (PB180120; Procell, Wuhan, China) at 37 °C with 5% $CO_2$.

The relevant small interfering RNAs against *CXCL3* (target seq: 5′-AGCTTATCA GCGTATCATTGACA-3′) and the control small interfering RNAs were synthesized by

**Table 2 Sequences of primers in qPCR.**

| Gene | Forward Primer (5′-3′) | Reverse Primer (5′-3′) |
|---|---|---|
| KCNK1 | ACTTCACCTCCGCGCTCTTCTT | AACAGGAGGGTGAAGGGAATGC |
| LIF | AGATCAGGAGCCAACTGGCACA | GCCACATAGCTTGTCCAGGTTG |
| TCN2 | CAGAACAGTGCGAGAGGAGATC | TCGCCTTGAGACATGCTGTTCC |
| SERPINF2 | AGACACCGTGTTGCTTCTCCTC | TGAACTGCTCGTCCAGGTGGAA |
| CXCL3 | TTCACCTCAAGAACATCCAAAGTG | TTCTTCCCATTCTTGAGTGTGGC |
| PIH1D2 | CTGTGTCAGTGGACAAGGATCC | GGCAACATCAATGACTGTGTAAGC |
| DTX1 | AGAATCCCGAGGATGTGGTTCG | TCGTAGCCTGATGCTGTGACCA |
| GCNT2 | TCCTGGTCCAAGGACACCTACA | CTGAGGTTTCCAGTCCAGGATG |
| GAPDH | GTCTCCTCTGACTTCAACAGCG | ACCACCCTGTTGCTGTAGCCAA |

GenePharma (Shanghai, China), which were then transfected into Hela cells with the use of Lipofectamine 2000 transfection reagent (11668-030; Thermo Fisher, Waltham, MA, USA) as per the manuals of the producer. All cells were harvested 48 h later for subsequent studies.

## Reverse-transcription quantitative PCR

Each sample was run in three independent triplicates for the quantification assay. In detail, total cellular RNA from HCerEpiC and CC cells was extracted by RNAiso Plus total RNA extraction kit (9108; Takara Bio, Shiga, Japan) and subjected to the PCR analysis in the ABI7500 thermocycler (Thermo Fisher, Waltham, MA, USA). In detail, following the extraction of the RNA and the quantification on ND-2000 spectrophotometer (Thermo Fisher, Waltham, MA, USA), the relevant cDNA was prepared from 1 µg total RNA and synthesized with a commercial cDNA synthesis kit (6110A; Takara Bio, Kusatsu, Japan). The PCR was hereafter initiated using the PrimeScript™ RT reagent kit (RR037Q; Takara Bio, Kusatsu, Japan) in the thermocycler at the following parameters: 95 °C for 30 s × 1 cycle and (95 °C for 5 s and 60 °C for 30 s) × 40 repeated cycles. $2^{-\Delta\Delta CT}$ method was adopted to calculate the results, with GAPDH as a housekeeping control (*Livak & Schmittgen, 2001*; *Sindhuja, Amuthalakshmi & Nalini, 2022*). See Table 2 for the primers used.

## Transwell cell migration/invasion assay

The transfected CC cells were transferred to the upper Transwell chamber (pore: eight-micron meter, 3422; Corning Inc., Corning, NY, USA) added with 200 micron liter serum-free media coated with/out the matrigel matrix (C0371; Beyotime, Shanghai, China). The lower Transwell chamber was added with 700 micron liter complete media and then the cells were cultured for 48 h. Next, all cells migrated/invaded to the lower Transwell chamber were serially fixed and dyed for 20 min, followed by the observation in three randomly picked areas in a light microscope (DP27, Olympus, Tokyo, Japan) (*Zhang et al., 2023*).

## Statistics

The statistical analyses and the data visualization were realized in R software version 3.6.0 (*R Core Team, 2019*) and GraphPad Prism software version 8.0.2. Difference on the variables between two groups were compared using Wilcoxon test or student's $t$ test. The correlation was calculated based on Spearman's correlation test. Logarithmic rank test was applied in the KM survival analysis. Statistically significant difference was defined when the $p$-value was less than 0.05 (In the figures, ns: $p > 0.05$; *: $p < 0.05$, **: $p < 0.01$, ***: $p < 0.001$ and ****: $p < 0.0001$).

# RESULTS

## Recognition of SiaTs-relevant biomarkers *via* WGCNA

A total of 20 SiaTs have been identified so far and their abnormal expression pattern in CC was observed. Specifically, it was noted in 34 CC sample that 14 SiaTs were mutated, with *ST6GALNAC3*, *ST6GAL1* and *ST8SIA5* ranking among the first three (Fig. 1A). The SiaTs score was hereafter calculated and relevant SiaTs-relevant gene modules were recognized and established. The samples were then clustered and screened for co-expression modules. The soft threshold β was set to 6 so as to ensure the scale-free network (Fig. 1B). Next, the specific gene modules were recognized *via* hierarchical clustering, and 17 modules were generated following the merging process (Fig. 1C). Among these modules, the grey module was the one which failed to be aggregated into other modules. The number of genes in the modules was listed in Fig. 1D. The correlation between the gene module and SiaTs score was determined in the end, where a strong correlation was seen in between the black module and the SiaTs score (Fig. 1E).

## Enrichment analysis on genes in the black module

The ClusterProfiler R package was hereafter applied for the enrichment analysis on the genes in black module at the sorting threshold of $p$-value $< 0.05$. The relevant results were seen in Figs. 2A–2D. Based on the KEGG analysis, the genes were mainly enriched in metabolism-relevant pathways like Glycolysis/Gluconeogenesis, Glycosphingolipid biosynthesis-lacto and neolacto series and Alanine, Amino sugar and nucleotide sugar metabolism, aspartate and glutamate metabolism (Fig. 2A). The enrichment analysis on the relevant biological process has hinted the significant enrichment on organic anion transport, carboxylic acid catabolic process and O-glycan processing, small molecule catabolic process (Fig. 2B). Also, genes in the black module were observed in certain cellular components like ciliary part, apical part of cell as well as apical plasma membrane (Fig. 2C). In addition, the enrichment analysis on the molecular function has demonstrated the enrichment of these genes in oxidoreductase activity, acting on the CH-OH group of donors, NAD or NADP as acceptor, aspartic-type endopeptidase activity and aspartic-type peptidase activity (Fig. 2D).

## Development of a prognostic model and verification

The samples in TCGA-CESC were grouped at the ratio of 7:3 into the train cohort and the test cohort. Then the genes in the black module were sorted at the parameters of

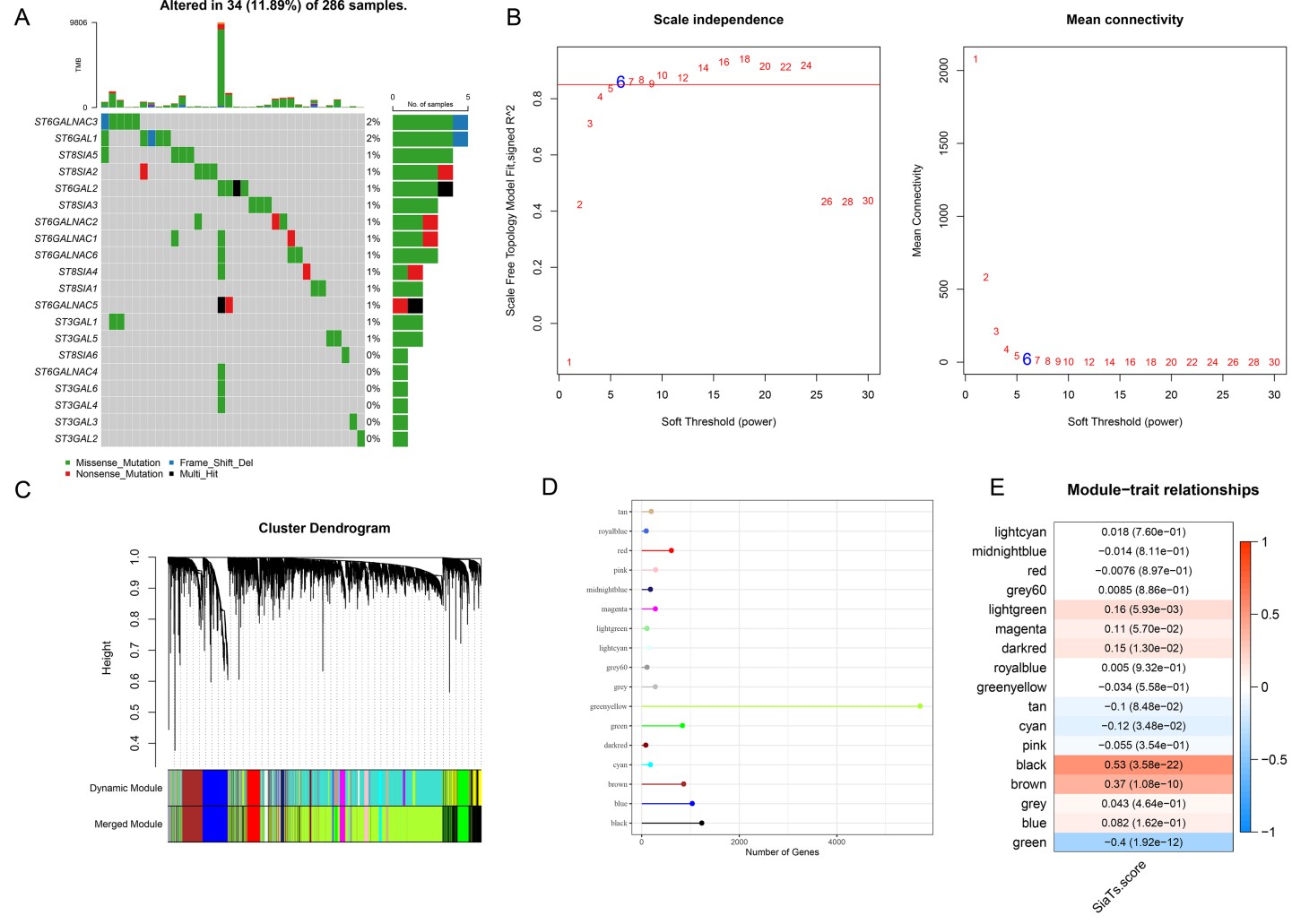

**Figure 1 Recognition of SiaTs-relevant biomarkers *via* WGCNA.** (A) Mutation status of SiaTs family members. (B) Scale-free fitting index analysis and mean connectivity analysis on sorting out the optimal soft threshold β. (C) Gene modules completion based on the dynamic tree cut method *via* hierarchical clustering on genes using 1-TOM as the distance measure. (D) Number of genes within the specific gene module. (E) Correlation between the gene modules and the SiaTs score.

*p*-value < 0.05 and correlation coefficient with SiaTs score > 0.3. A total of 445 genes were thereafter obtained and further narrowed down *via* Cox regression analysis based on the data in the training set at the threshold of *p*-value < 0.05, followed by the LASSO Cox regression analysis (Fig. 3A). Finally, eight genes were identified and applied for developing a prognostic model based on the following formula (Fig. 3B):

$$\begin{aligned} \text{RiskScore} = {}& 0.455 * KCNK15 + 0.26 * CXCL3 - 0.715 * PIH1D2 \\ & - 0.524 * DTX1 + 0.295 * LIF - 0.443 * TCN2 - 0.26 * SERPINF2 \\ & - 0.286 * GCNT2 \end{aligned}$$

Patients were then allocated to different risk groups by the optimal cut-off value and the good prognosis was seen in patients of the low-risk group in the training and validation cohorts as well as the TCGA cohort (Figs. 3C–3E). The AUC value was

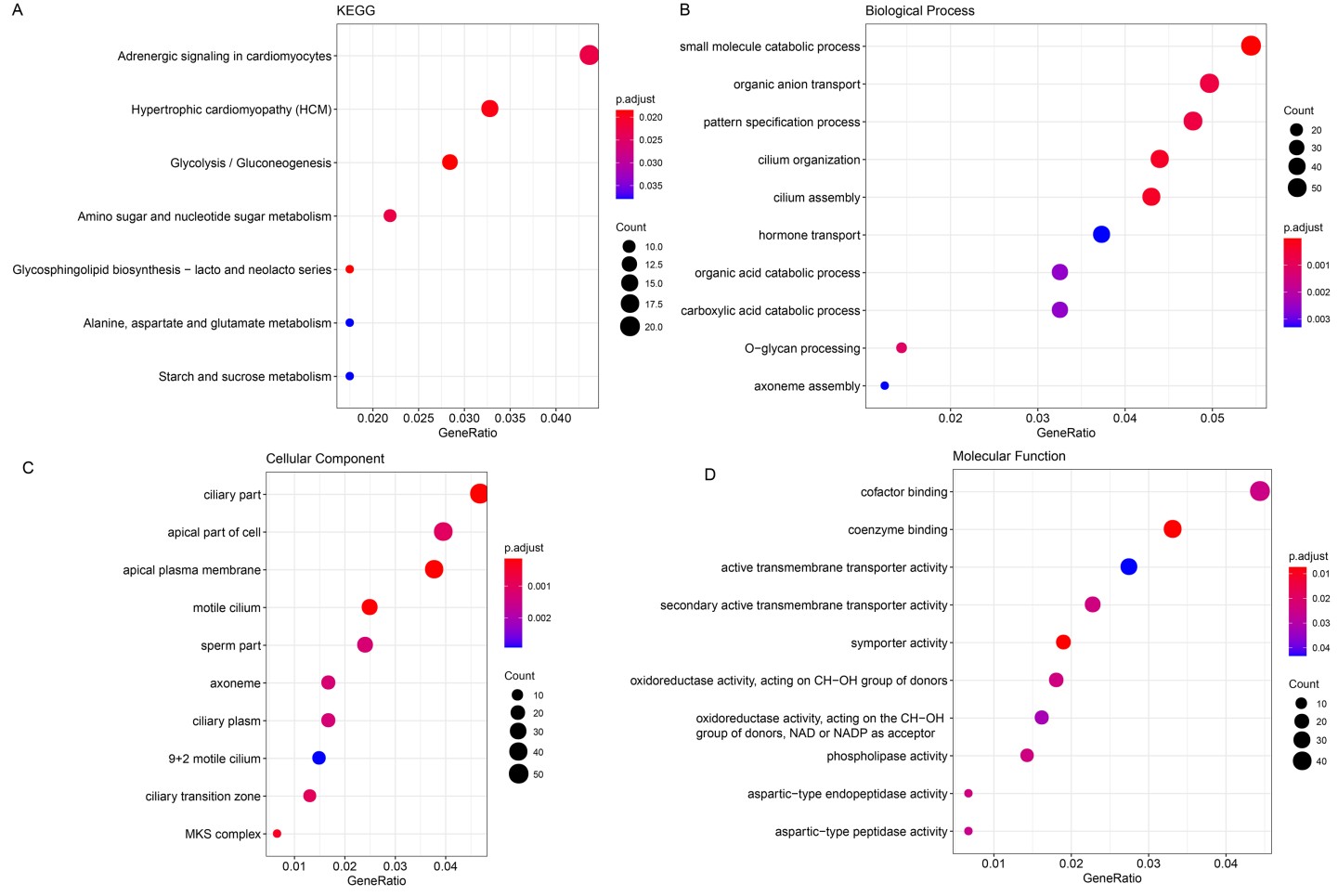

**Figure 2 Enrichment analysis on genes in the black module.** (A–D) Enrichment analysis on genes in the black module, based on KEGG enrichment analysis (A) and analyses on their enrichment in the relevant biological process (B), cellular component (C), and molecular function (D).

calculated to reflect the prediction accuracy. High AUC value was seen in the training and validation cohorts as well as the TCGA cohort (Figs. 3F–3H). It was also noticed that the percentage of mortality in patients of high-risk group was evidently higher, based on the Chi-square test (Fig. 3I).

To better characterize the model robustness, similar validation was applied in the dataset GSE44001. Notably, the low-risk group had a better survival (Fig. 3J) and a high AUC value (Fig. 3K).

## Immune cell infiltration in different risk groups

The immune cell infiltration was determined with ESTIMATE algorithm to compare the difference on the TIME in patients of different risk groups. A higher immune infiltration was seen in low-risk patients (Fig. 4A). With the purpose of exploring the correlation between the RiskScore and immune functions, the score of 6 types of immune cells was calculated by TIMER, discovering the higher score of B_cell, CD4_Tcells and Macrophages

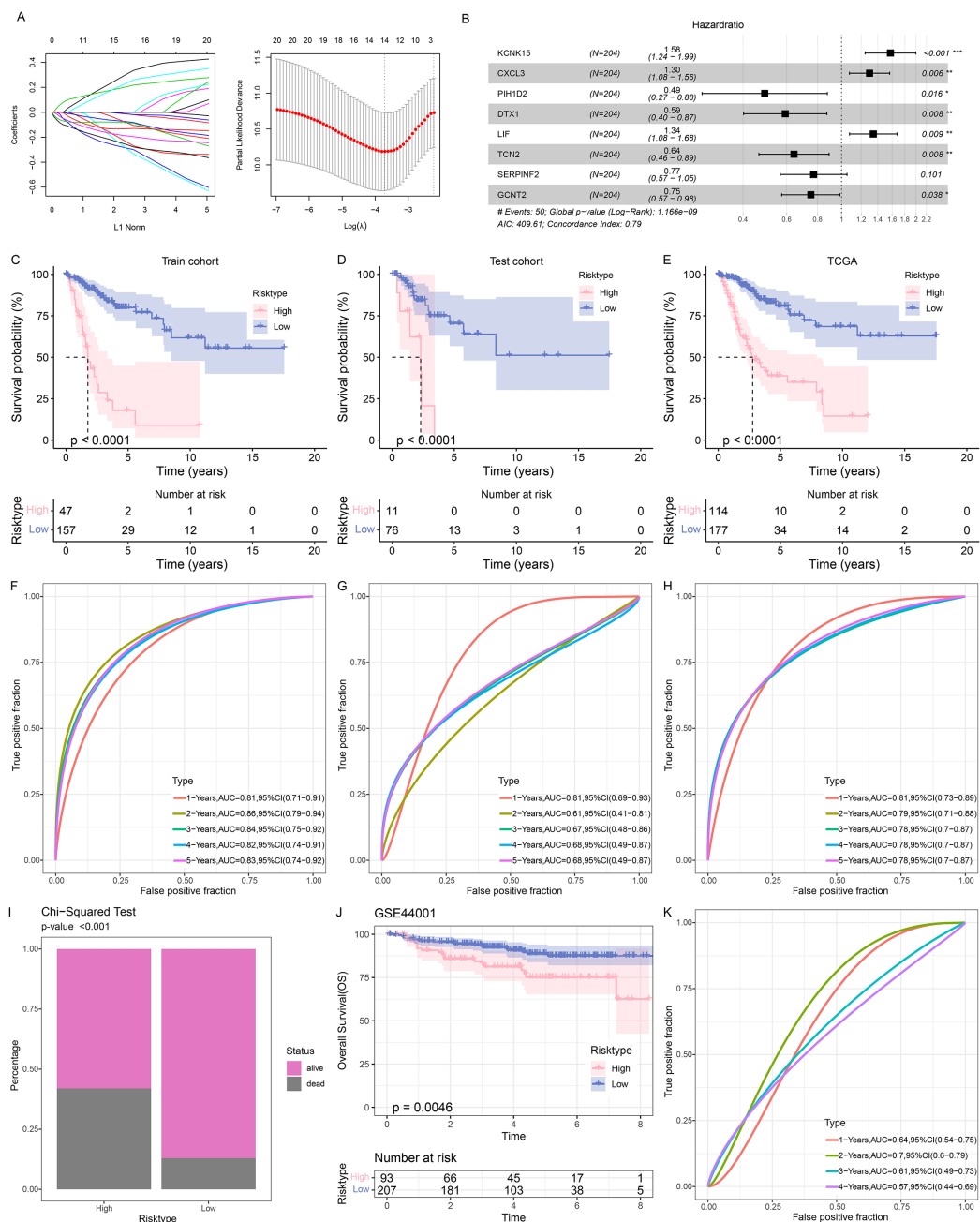

**Figure 3 Construction and validation of the prognostic model.** (A) Sorting on key prognostic genes *via* LASSO regression analysis. (B) Plot of multifactorial random forest analysis. (C–E) Kaplan-Meier survival analysis on the train cohort (C), test cohort (D) and TCGA cohort (E). (F–H) ROC curve of the RiskScore based on the train cohort (F), test cohort (G) and TCGA cohort (H). (I) Survival status in patients of different risk groups based on TCGA cohort. (J) Kaplan-Meier survival analysis on the GSE44001 dataset. (K) ROC curve of the RiskScore based on the GSE44001 dataset. *: $p < 0.05$, **: $p < 0.01$, ***: $p < 0.001$.

in the low-risk group (Fig. 4B). This means that low-risk patients had a stronger anti-tumor immune response. Furthermore, the association between RiskScore and the immune cell score was also calculated and determined by MCP-counter method. Visibly,

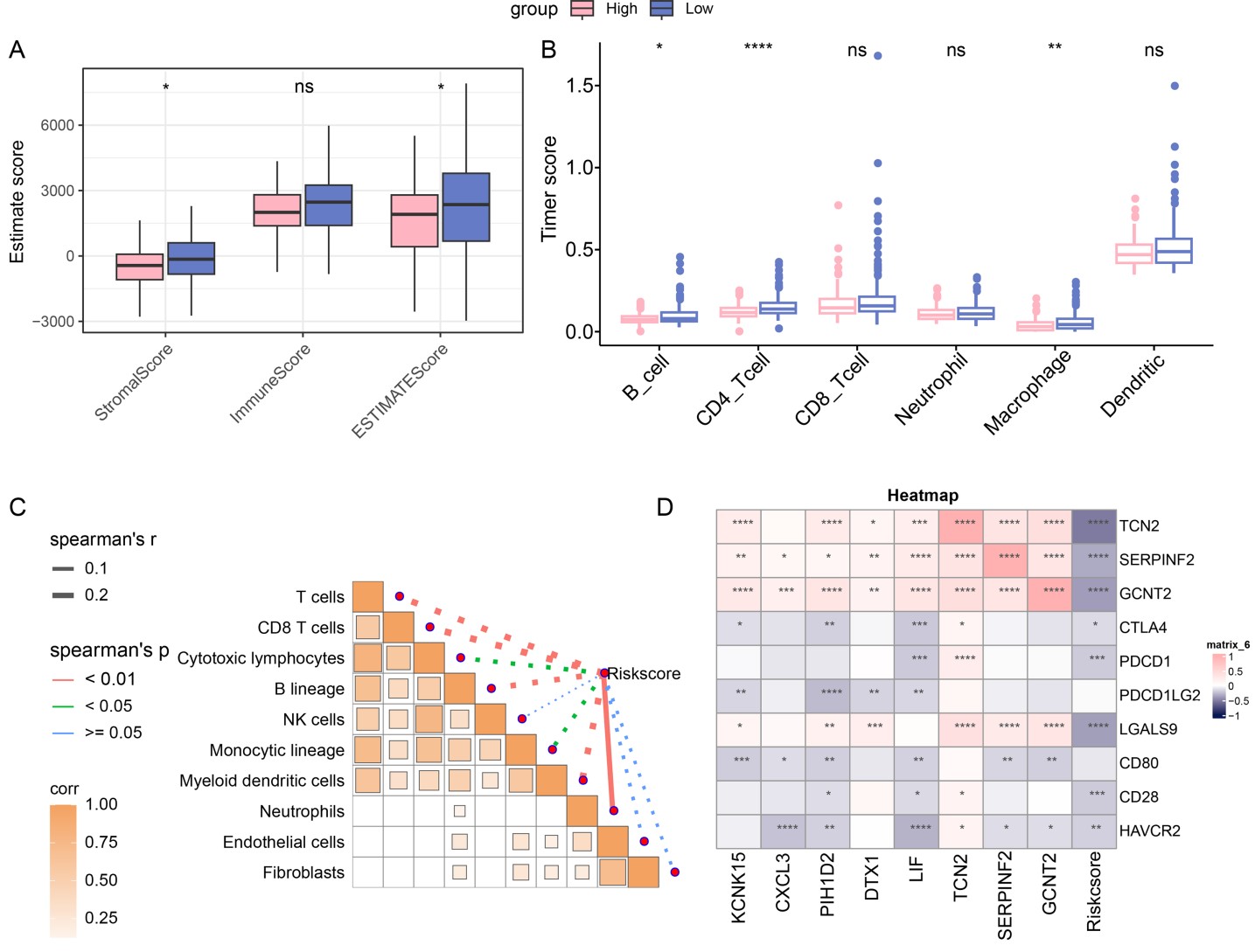

**Figure 4 Immune cell infiltration in different risk groups.** (A) Immune infiltration in different risk groups determined *via* ESTIMATE algorithm. (B) TIMER was applied to calculate the immune infiltration in different risk groups. (C) Correlation between immune cells score and RiskScore, determined *via* MCP-counter method. The solid lines in the figure represent positive correlations and the dashed lines represent negative correlations. (D) Correlation in RiskScore and prognosis- and immune checkpoint-related genes. (ns: $p > 0.05$; *: $p < 0.05$, **: $p < 0.01$, ***: $p < 0.001$ and ****: $p < 0.0001$).

the RiskScore was negatively correlated with NK cells, T cells, cytotoxic lymphocytes, monocytic lineage, CD8 T cells, myeloid dendritic cells, and B lineage (Fig. 4C), hinting the higher RiskScore was indicative of a lower immune infiltration status. In addition, the correlation in RiskScore and prognosis- and immune checkpoint-related genes were evaluated, and it was noted that RiskScore was negatively linked to the following genes including *HAVCR2*, *CD28*, *LGALS9*, *PDCD1* and *CTLA4* (Fig. 4D). In conclusion, these results indicate that patients in the low-risk group had higher levels of immune cell infiltration and stronger immune responses, suggesting that their immune microenvironment may be more active.

### Abnormality in gene mutation and relevant pathways in different risk groups

The gene mutation status in different risk groups was investigated, and the top 10 genes with relatively higher mutation frequency in these two groups were listed in Figs. 5A,5B. It was seen that *TTN* and *PIK3CA* ranked the first and second in both risk groups, while *KMT2C* and *MUC16* occupied the third position in the high- and low-risk groups, respectively.

The ssGSEA was then initiated using TCGA dataset to determine the relevant pathways in different risk groups. The ssGSEA score of each sample in HALLMARK Pathway was calculated and the correlation between the RiskScore and the calculated score was additionally analyzed. It was noted that the RiskScore was positively correlated with MYC Targets V2 and TGF Beta Signaling yet negatively correlated with Interferon Gamma Response (Fig. 5C), thus hinting that higher RiskScore may be associated with the promoted growth of tumor cells and the suppressed immunoreactivity.

Next, the differential activated pathways in different risk groups were compared and analyzed. The dataset h.all.v2023.1.Hs.entrez.gmt was applied for GSEA and the significant enrichment was defined by the threshold FDR < 0.05. Relevant results have demonstrated that Angiogenesis and MYC target V2 may be the differentially activated pathways in high-risk patients, while those in low-risk patients were KRAS Signaling DN and Interferon alpha response (Figs. 5D,5E).

### Difference on the immuno/chemotherapy response in different risk groups

First, we compared the different immunotherapy response and applied TIDE software to assess the potential efficacy of immunotherapy in different risk groups. It was uncovered in Fig. 6A that insignificance on the TIDE score was seen in the two groups, whilst the lower Exclusion score and higher MDSC score were observed in the high-risk group than those in the low-risk group. Meanwhile, we also determined the difference on the expression levels of immune checkpoint genes in these groups, and it was unveiled that most genes had a high expression in the low-risk group, including *CTLA4*, *HAVCR2*, *LAG3*, *TIGIT*, and *PDCD1* (Fig. 6B). Furthermore, the response to the chemotherapy was analyzed in different risk groups of TCGA-CESC cohort based on the estimated IC50 value, and the lower IC50 values of paclitaxel, cisplatin, crizotinib, erlotinib, sunitinib, and sorafenib were seen in high-risk group in comparison with those in low-risk group (Fig. 6C). These results indicate that low-risk patients with CC have higher levels of immune infiltration and more active levels of immune checkpoint expression, suggesting a starting point more amenable to immunotherapy, while high-risk patients are more likely to benefit from chemotherapy due to the presence of more immunosuppressive factors.

### Quantification on the SiaTs-related genes in CC cells

For subsequent cellular validation, we firstly measured the expressions of all 8 SiaTs-related genes in CC cells Hela and HCerEpic cells. Relevant results have revealed the differential expressions of these genes in CC cells Hela, where the higher expression levels

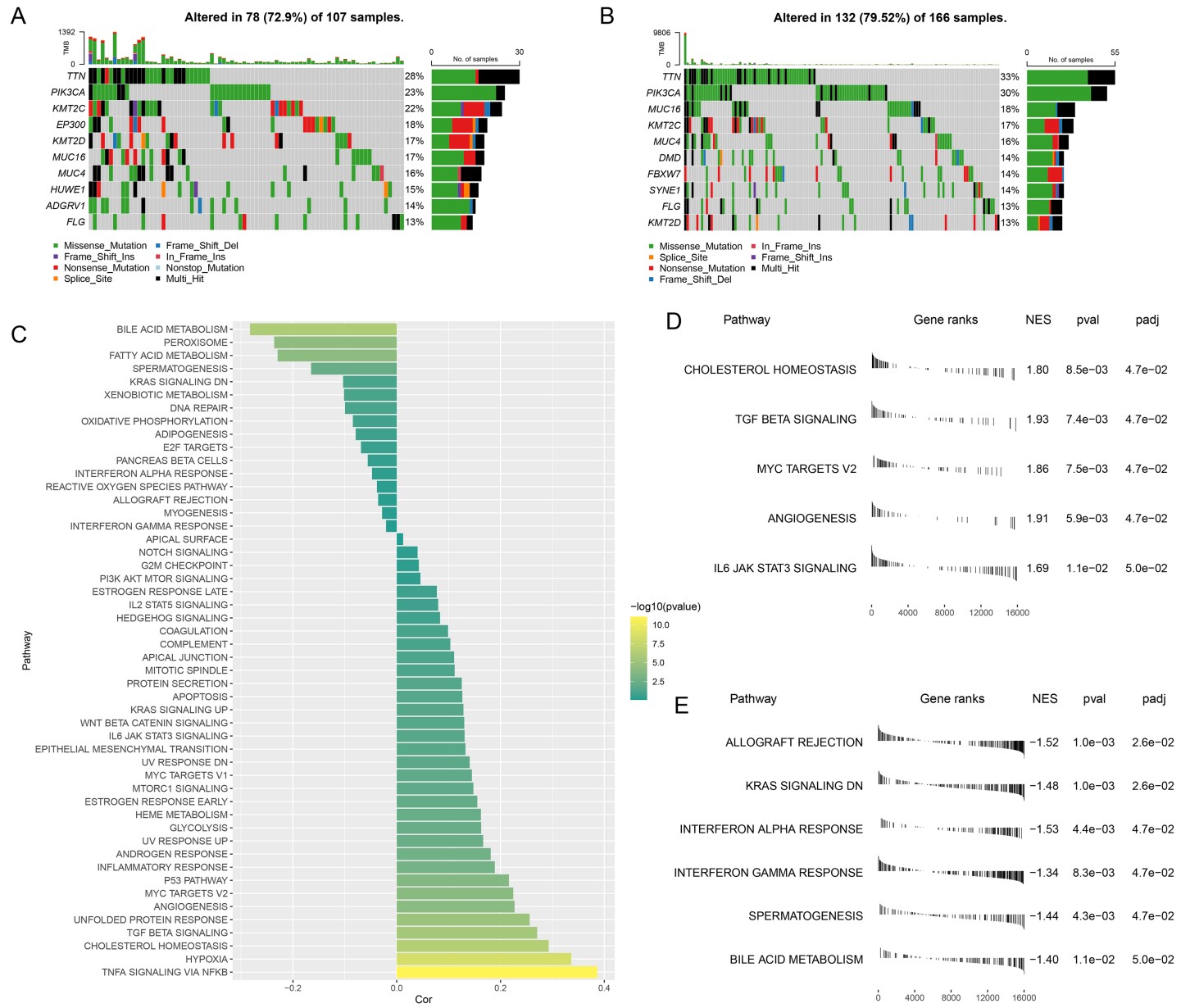

**Figure 5 Abnormality in gene mutation and relevant pathways in different risk groups.** (A,B) Top 10 genes with relatively higher mutation frequency in high-risk (A) and low-risk groups (B). (C) Correlation between the RiskScore and the calculated ssGSEA score of each sample in HALLMARK Pathway. (D,E) GSEA analysis on the differential activated pathways in high-risk (D) and low-risk groups (E).

of *KCNK15*, *LIF*, *TCN2*, *SERPINF2*, and *CXCL3* and the lower expression levels of *PIH1D2*, *DTX1* and *GCNT2* were observed in comparison with those in HCerEpic cells (Fig. 7A, $p < 0.05$).

*CXCL3* has been reported to play a key role in tumor development and progression (*Cheng et al., 2023*). To further investigate its potential function in CC, we designed specific small interfering RNAs and transfected them into CC cells as needed. The *in-vitro* migration and invasion experiments of the effects of *CXCL3* silencing on the CC cells

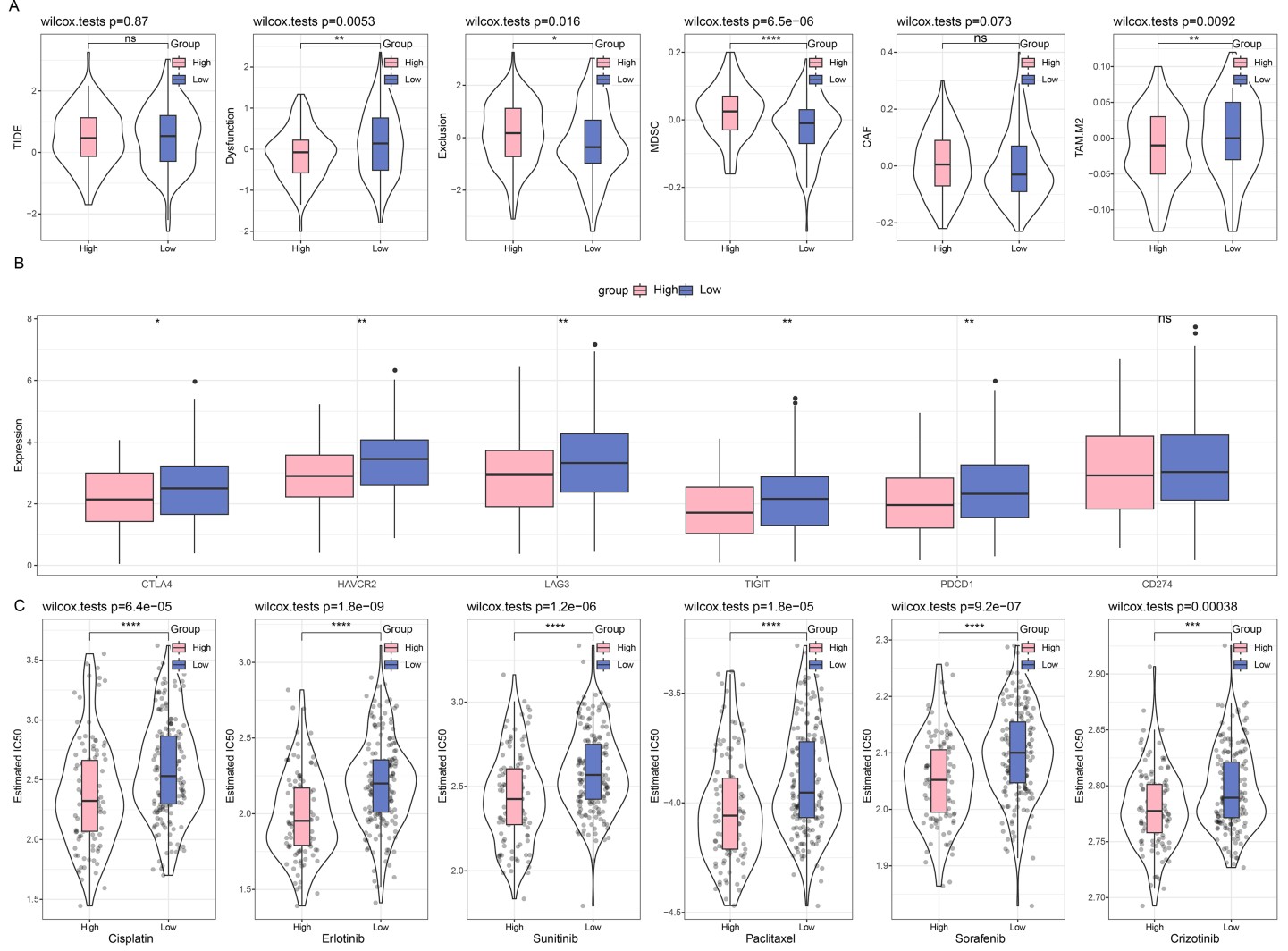

**Figure 6 Difference on the immuno/chemotherapy response in different risk groups.** (A) TIDE score on the different risk groups of TCGA-CESC cohort. (B) Levels of immune checkpoint genes in different risk groups of TCGA-CESC cohort. (C) The box plots of the estimated IC50 value for Cisplatin, Erlotinib, Sunitinib, Paclitaxel, Sorafenib, and Crizotinib in different risk groups of TCGA-CESC. (ns: $p > 0.05$; *: $p < 0.05$, **: $p < 0.01$, ***: $p < 0.001$ and ****: $p < 0.0001$).

showed that *CXCL3* silencing could visibly reduce the number of migrated and invaded Hela cells at 48 h (Figs. 7B, 7C, $p < 0.001$).

## DISCUSSION

CC ranks the fourth in the most prevalent female malignancies, which has posed a great threat and challenge around the globe (*Momenimovahed et al., 2023*). Accumulating evidences have stressed the vital importance of PTM in the progression of cancers, including CC (*Ju et al., 2020*). When it comes to glycosylation in CC, the fundamental role of glyciobiology has been already illustrated, which is believed to be relevant with the alternation in the expression of N-linked glycan and the abnormality on the expression of terminal glycan structures (*Xu et al., 2021*). Further, while examining the specific

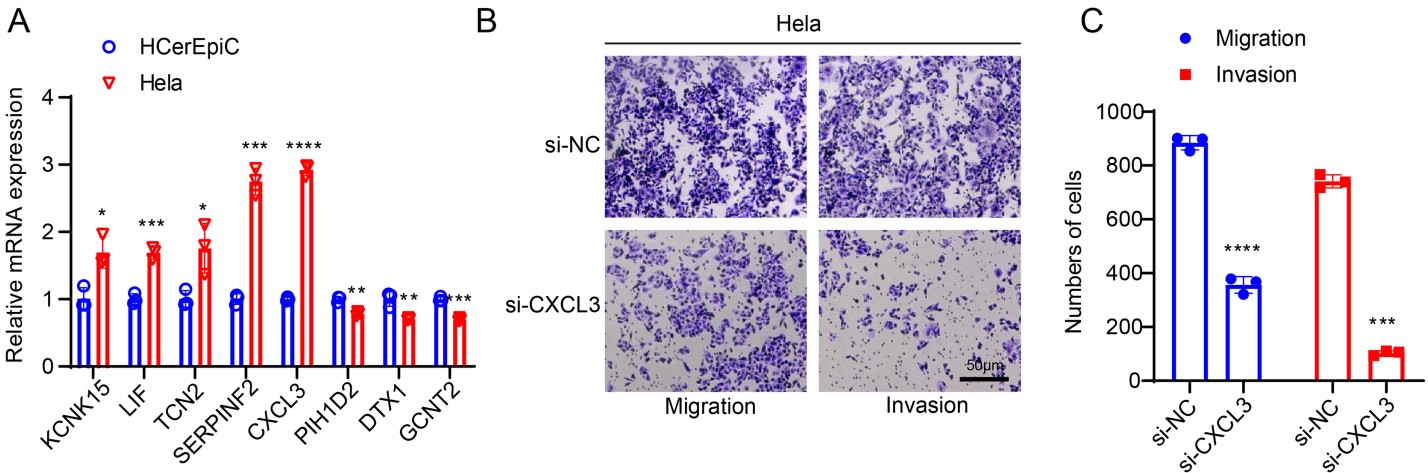

**Figure 7** *In-vitro* **cell validation results.** (A) Quantification on the SiaTs-related genes in CC cells Hela and human cervical endometrial epithelial cells (HCerEpiC) *via* reverse-transcription quantitative PCR. (B,C) Effects of *CXCL3* silencing on the migration and invasion of CC cells Hela at 48 h. All results from three independent triplicates were expressed as mean ± standard deviation. (*: $p < 0.05$, **: $p < 0.01$, ***: $p < 0.001$ and ****: $p < 0.0001$).

involvement of SiaTs in CC, it has been suggested that ST3Gal IV can mediate the in-vitro and in-vivo growth of CC cells (*Wu et al., 2020*). Also, the knockdown of ST6Gal-I can sensitize CC cells to cisplatin (*Zhang et al., 2016*). In our current study, based on an existing study highlighting the SiaTs-related gene signature as a potential predictor of therapeutic response and prognosis in bladder cancer (*Cao et al., 2023*), further explore the involvement of SiaTs-related gene signature in CC. Relevant results have revealed the SiaTs-relevant genes were mainly enriched in pathways related to metabolism. The established RiskScore model based on these relevant genes manifested excellent prognostic classification efficiency. These results have additionally proven the efficacy of SiaTs-related gene signature to predict the therapeutic response for prognosis of CC.

In our study, the RiskScore model applied to stratify patients into different risk groups was established in accordance with those relevant to SiaTs using a series of in-silico analyses like WGCNA, Lasso regression analysis and Cox regression analysis, which encompasses *KCNK15*, *CXCL3*, *PIH1D2*, *DTX1*, *LIF*, *TCN2*, *SERPINF2* and *GCNT2*. *KCNK15* has been already explored in other female malignancy like triple-negative breast cancer where its underexpression is evidently related to the phenotype (*Dookeran et al., 2017*). *CXCL3* is a member of the ELR$^+$ CXC subfamily of the chemokine CXC family which is overexpressed in tumors and exerts a promoting effect on the tumorigenic potential of CC cells (*Reyes et al., 2021*; *Qi et al., 2020*). *PIH1D2* was previously screened out in the univariate Cox, random survival forest, and multivariate Cox regression analysis to be a RNA-binding protein gene in renal papillary cell carcinoma (*Jiang et al., 2021*). *DTX1* belongs to the human DTX protein family (the putative E3 ubiquitin ligases) which has been indicated to be a potential tumor suppressor gene based on an integrative computational analysis on both transcriptional and epigenetic alternations in head and neck squamous cell carcinoma (*Scalia et al., 2023*; *Gaykalova et al., 2017*). *LIF* is a

constituent of the IL-6 cytokine family and can repress the proliferation of CC cells (*Jorgensen & de la Puente, 2022*; *Bay, Patterson & Teng, 2011*). *TCN2* was known as both a Vitamin B (*Scott et al., 2023*) transporter protein and a biomarker associated with the risk of thyroid cancer in a multi-omics Mendelian randomization study (*Zhang et al., 2017*; *Sun et al., 2024*). Alternatively called α2-antiplasmin, *SEPINF2* refers to a gene which can not only encode plasmin inhibitors degrading plasma fibrin and other proteins but also act as a good predictor of overall survival in breast cancer (*Franceschini et al., 2012*; *Verbree-Willemsen et al., 2020*; *Barrón-Gallardo et al., 2022*). *GCNT2* is one of the major glycosyltransferases applied for the diagnosis and staging classification on colon cancer (*Barrón-Gallardo et al., 2022*; *Perez et al., 2021*). In this study, these eight genes have been screened to be relevant to and correlated with SiaTs in CC, and a relevant RiskScore model applying these six genes has demonstrated a strong potential in determining the prognosis of CC patients, based on the data from both TCGA-CESC and the dataset GSE44001, thus hinting that these genes as relevant targets could be applied and examined in clinical practice.

The critical role of immune infiltration in the prediction of both immune escape and immunotherapy in CC has been already addressed and interpreted, as exemplified by the discovery suggesting that higher immune infiltration score was indicative of a stronger tendency to develop immune escape (*Chen et al., 2023*). In our study, notably, the RiskScore was negatively correlated with NK cells, monocytic lineage, T cells, CD8 T cells, cytotoxic lymphocytes, myeloid dendritic cells, and B lineage. NK cells are capable of destroying tumor cells that showcase surface markers linked to oncogenic changes, exhibit death ligands, or demonstrate reduced levels of MHC molecules. Furthermore, their capacity for antibody-dependent cytotoxicity, along with their ability to generate inflammatory cytokines—which work in conjunction with the activation of additional cytotoxic immune cells—equips NK cells to function as effective anticancer agents (*Gutiérrez-Hoya & Soto-Cruz, 2021*). A previous study applying single-cell RNA sequencing on CC has revealed the abundance of immune cells like plasma cells, mast cells, NK/T cells, macrophages, B cells, neutrophils in different stages of CC (*Su et al., 2022*). Another study has revealed the heterogeneity in malignant cells of CC, where cancer-associated fibroblasts and CD8[+] T cells stand out (*Li et al., 2022*). In addition, *Shen et al. (2022)* explored genes linked to CD8+ T-cell infiltration in cancer and demonstrated that CD8[+] T-cell infiltration was the only beneficial and independent factor in influencing cancer prognosis. These phenomena have important immunological implications in that these cells play a central role in the antitumor immune response, especially NK cells and CD8[+] T cells, whose direct killing role is a key mechanism for limiting tumor growth and metastasis. Overall, the characterization of SiaTs-related genes revealed in this study not only contributes to the optimization of stratified management and therapeutic decision-making for CC patients, but also suggests that these genes may have potential biological functions in regulating the immune microenvironment, influencing the intensity of immune infiltration and anti-tumor immune activity.

This study systematically constructed a prognostic risk model centered on SiaT-related genes. By integrating immune microenvironment analysis and drug sensitivity prediction,

we demonstrated that this model can effectively guide immunotherapy and chemotherapy strategies for CC patients, thereby providing a novel theoretical foundation and practical framework for precise stratified therapy and personalized medicine. However, some limitations of the study remain. First, this study primarily relies on transcriptomic and clinical data of CC patients from public databases for model construction and validation. Although the sample size is relatively large, potential biases—such as single-source data, uneven geographical distribution of samples, and incomplete clinical information—may affect the model's generalizability and clinical applicability. To address these limitations, we plan to collaborate with multiple centers to collect independent validation cohorts from diverse populations and regions, thereby improving the model's robustness and generalizability. Additionally, we will incorporate clinical characteristics, disease stages, treatment regimens, and other factors into multivariate analyses to enhance the model's clinical interpretability. Second, our findings still require further validation of the specific role of SiaTs-related genes in CC using mouse tumor models or other *in vivo* experiments. Finally, this study focused on analyzing some SiaTs-related genes but did not cover all SiaTs family members. This may cause an incomprehensive understanding of the role of the SiaTs family in CC, overlooking the potential role of other SiaTs family members in tumor progression and immune regulation. Therefore, subsequent studies will expand the range of SiaTs family members, systematically analyze their expression characteristics and functional roles in various CC subtypes, pay special attention to understudied rare members, and explore their value as potential therapeutic targets or immunomodulatory factors.

## CONCLUSION

This study systematically analyzed the immune microenvironment and drug sensitivity in cervical cancer patients, revealing the critical role of SiaTs-related gene signatures in predicting prognosis and treatment response. Immune microenvironment analysis indicated that high-risk patients tend to exhibit stronger immune evasion, while low-risk patients show higher immune infiltration and immune checkpoint gene expression levels. Drug sensitivity analysis demonstrated that high-risk patients are more sensitive to chemotherapy drugs such as cisplatin and sorafenib. These findings offer new perspectives for personalized therapy, especially in combining immunotherapy and chemotherapy. The risk score model developed in this study provides valuable guidance for risk stratification and treatment decision-making in clinical practice, advancing personalized medicine.

## ABBREVIATION

| | |
|---|---|
| **CC** | Cervical cancer |
| **HPV** | human papillomavirus |
| **PTM** | post-translational modification |
| **SiaTs** | sialyltransferases |
| **TCGA** | the cancer genome atlas |
| **WGCNA** | Weighted gene co-expression analysis |
| **GO** | gene ontology |

| KEGG | Kyoto Encyclopedia of Genes and Genomes |
|---|---|
| ROC | receiver operator characteristic |
| AUC | area under the curve |
| TIME | Tumor immune microenvironment |
| TIDE | Tumor immune dysfunction and exclusion |
| HCerEpiC | human cervical endometrial epithelial cells |
| IC50 | half maximal inhibitory concentration |

### Funding

This project was supported by the Nantong Science and Technology Plan Project (No. JC22022107) and Special scientific research project of Nantong Health and Wellness Committee (No. MS2024047). The funders had no role in study design, data collection and analysis, decision to publish, or preparation of the manuscript.

### Grant Disclosures

The following grant information was disclosed by the authors:
Nantong Science and Technology Plan Project: JC22022107.
Special scientific research project of Nantong Health and Wellness Committee: MS2024047.

### Competing Interests

The authors declare that they have no competing interests.

### Author Contributions

- Jia Shao conceived and designed the experiments, performed the experiments, analyzed the data, authored or reviewed drafts of the article, and approved the final draft.
- Can Zhang performed the experiments, authored or reviewed drafts of the article, and approved the final draft.
- Yaonan Tang conceived and designed the experiments, analyzed the data, prepared figures and/or tables, and approved the final draft.
- Aiqin He performed the experiments, prepared figures and/or tables, and approved the final draft.
- Xiangyan Cheng conceived and designed the experiments, performed the experiments, analyzed the data, authored or reviewed drafts of the article, and approved the final draft.

### Data Availability

The public dataset used in this study is available in GSE44001.
The raw data is available at GitHub and Zenodo:
- https://github.com/jiashao123438/all-data.git.

- jiashao123438. (2025). jiashao123438/all-data: all data (v.1.1.0). Zenodo. https://doi.org/10.5281/zenodo.14829613.

## Supplemental Information

Supplemental information for this article can be found online at http://dx.doi.org/10.7717/peerj.19422#supplemental-information.

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
