# Peer review of "Sialyltransferase-related genes as predictive factors for therapeutic response and prognosis in cervical cancer"

_PeerJ, doi:10.7717/peerj.19422_

## Round 0.1 · original submission · Major Revisions

Based on the comments from two reviewers, while they recognize the significance of your work, several important concerns need to be addressed. Therefore, I would recommend a major revision of your manuscript. Please submit your revised version along with a point-by-point response to the reviewers' comments.

Reviewer 1 ·

Basic reporting

The main objective of this study was to identify signature genes related to prognosis and treatment response in cervical cancer by bioinformatics means, and silanesyltransferases (SiaTs) were selected as cancer progression-related features. This study presents a compactness in the idea; first, cervical cancer (CC)-related data were obtained from public databases, and gene modules related to SiaTs were obtained based on weighted gene co-expression network construction. Then, this study applied COX regression analysis to build a CC risk assessment model and combined algorithms such as ESTIMATE to reveal the predictive role of this model in the immune microenvironment, and combined multiple bioinformatics tools to evaluate the association between immunotherapy response and the risk assessment model. In conclusion, the idea of this study is relatively conventional, but the following issues still need to be addressed before publication:
1. Did the study acquire the feature of SiaTs as an important characteristic of CC progression only through systematic literature analysis and preliminary research? Please add the relevant survey analysis and the corresponding literature in the introduction section to confirm the distinctiveness of this paper's intention.
2. The analysis of TME in this study was mainly achieved by the ESTIMATE algorithm and TIMER, but do the metrics derived from these two methods cover most of the characteristics of TME? For example, can they cover most immune cell types and stromal cell types? It is recommended that this be clarified and added to the analysis where necessary.
3. Currently, several studies have reported novel immune checkpoints for CC treatment, whereas the focus of the analysis in this study remained on traditional immune checkpoints, thus suggesting the addition of an analysis on CC immune checkpoints in conjunction with the most recent reports.
4. The description of the results in Figure 6 is too general and does not indicate what conclusions are revealed by the differences in immune checkpoint genes and TIDE scores, and thus, it is suggested to indicate whether the differences in TIDE scores and immune checkpoint genes between samples can be tapped into as potential drugs or potential therapeutic strategies for CC treatment?
5. In this paper, CXCL3 was selected for the construction of genetically engineered cell lines, is it because of the high relative expression level of this gene in cancer cells? Please clarify this in the results section and elucidate the important role of CXCL3 in the development of CC in conjunction with existing reports.
6. The discussion section elucidates the critical role of immune infiltration in predicting CC immune escape and immunotherapy, but the subsequent description does not expand upon the key conclusions of the paper in conjunction with the focus of the paper, i.e., it does not indicate which cells in the immune microenvironment are practically regulated by the genes revealed herein, and it is suggested that additional clarification be provided.
7. Can the prognostically relevant biomarkers screened in this paper practically influence the heterogeneity of CC and thus immunotherapy? Please provide a systematic explanation in the Discussion section in conjunction with the main findings of this paper.
8. The limitations of this study need to be further described and added to the discussion section. It is recommended that the limitations of this paper be added in terms of the limitations of public databases, ideas for follow-up experiments, and current experimental samples that need to be added.
9. It is recommended that a summary description be added at the end of the discussion section of this paper to systematically summarize how the main findings of this paper (e.g., the findings of the immune microenvironmental analysis and the drug susceptibility analysis) contribute to the existing healthcare field, thus giving more insight into the landscape of this paper.
10. From the analysis of this study, RiskScore seems to correlate with pathways related to cellular metabolic reprogramming, such as fatty acid metabolism, and thus is it possible to discuss the conclusions of this paper from the perspective of cellular energy metabolism? Please give a reasonable explanation and add relevant literature if necessary to show that the biomarkers mined in this paper can actually affect metabolic reprogramming in cancer cells.

Experimental design

no comment

Validity of the findings

no comment

Reviewer 2 ·

Basic reporting

no comment

Experimental design

no comment

Validity of the findings

no comment

Additional comments

In this study, various bioinformatics techniques such as WGCNA and unifactorial and multifactorial analyses were applied to obtain eight biomarkers and construct a risk model, performed enrichment analysis, immune infiltration analysis, and drug susceptibility analysis on the key genes we obtained, and finally verified the role of these genes on CC using in vitro experiments. The following changes are needed for this study:
1. What exactly are the cell experiments referred to in the methods section of the abstract? Please describe in detail to prevent confusion for the reader.
2. The names of the genes involved need to be in italics and the proteins in block letters.
3. Abbreviations that appear for the first time in the text need to be given in full, e.g., TCGA, TPM, etc., and the same elsewhere.
4. What are the meanings of the dotted and solid lines in Figure 4C, and the related diagrammatic notes are not clear enough.
5. Why was CXCL3 chosen for the study of cancer cell migration and invasion?
6. Figure 3's F-H characters are out of bounds and need to be adjusted.
7. Why did you split the TCGA-CESC samples into training and validation sets in the ratio of 7:3 and use GSE44001 as the new validation set in Figure3? Are there any characteristics of the samples in these two datasets?
8. KCNK1 in Figure 7 is misspelled and should be KCNK15.
9. What is the reason for the elevated expression of the protective gene SERPINF2 in Figure7 in the Hela group?
10. Patients in the low-risk group have higher levels of immune cell infiltration and stronger immune cell infiltration ability does this imply that the high- and low-risk groups are suitable for different treatment modalities. Immunotherapy is one of the important treatments for cancer nowadays, and it is suggested to add a discussion about it in the discussion section.
11. Similarly, this study also compares differences in immune/chemotherapy response in different risk groups, which has implications for stratification of treatment, and in addition, the title of this study is “Revelation and recognition of a genes signature related to sialyltransferases for evaluating therapeutic response and prognosis in cervical cancer” so the addition of the relevant content in the discussion section could enhance the link to the title.

---

## Round 0.2 · accepted · Accept

Based on the positive feedback from both reviewers and your careful revision of the manuscript, I am pleased to inform you that your paper has been accepted for publication.

Reviewer 1 ·

Basic reporting

no comment

Experimental design

no comment

Validity of the findings

no comment

Additional comments

The main objective of this study was to identify signature genes related to prognosis and treatment response in cervical cancer by bioinformatics means, and silanesyltransferases (SiaTs) were selected as cancer progression-related features. This study presents a compactness in the idea; first, cervical cancer (CC)-related data were obtained from public databases, and gene modules related to SiaTs were obtained based on weighted gene co-expression network construction. Then, this study applied COX regression analysis to build a CC risk assessment model and combined algorithms such as ESTIMATE to reveal the predictive role of this model in the immune microenvironment, and combined multiple bioinformatics tools to evaluate the association between immunotherapy response and the risk assessment model. The author has responded to the previous comments and the current version is complete. Although there is room for improvement in innovation, the workload is sufficient.

Reviewer 2 ·

Basic reporting

In this study, various bioinformatics techniques such as WGCNA and unifactorial and multifactorial analyses were applied to obtain eight biomarkers and construct a risk model, performed enrichment analysis, immune infiltration analysis, and drug susceptibility analysis on the key genes we obtained, and finally verified the role of these genes on CC using in vitro experiments. The author provided a detailed response to my question, and I do not have any new comments now

Experimental design

no comment

Validity of the findings

no comment